# Differential Expression Profiles of Oxidative Stress Levels, 8-oxo-dG and 4-HNE, in Barrett’s Esophagus Compared to Esophageal Adenocarcinoma

**DOI:** 10.3390/ijms20184449

**Published:** 2019-09-10

**Authors:** Naoimh J. O’Farrell, James J. Phelan, Ronan Feighery, Brendan Doyle, Sarah L. Picardo, Narayanasamy Ravi, Dermot O’Toole, John V. Reynolds, Jacintha O’Sullivan

**Affiliations:** 1Trinity Translational Medicine Institute, Department of Surgery, Trinity College Dublin, St. James’ Hospital, Dublin 8, Ireland; 2Department of Histopathology, St. James’ Hospital, Dublin 8, Ireland; 3Department of Clinical Medicine, Trinity Translational Medicine Institute, St. James’ Hospital, Dublin 8, Ireland

**Keywords:** mitochondrial instability, oxidative stress, Barrett’s esophagus, esophageal adenocarcinoma, 8-oxo-dG, 4-HNE

## Abstract

Barrett’s esophagus (BE), a chronic inflammatory condition, is the leading risk factor for esophageal adenocarcinoma (EAC). In inflammation to cancer pathways, oxidative stress profiles have been linked to cancer progression. However, the relevance of oxidative stress profiles along the BE-disease sequence remains to be elucidated. In this study, markers of oxidative stress; DNA adducts (8-oxo-dG) and lipoperoxidation (4-HNE), and markers of proliferation (Ki67) were measured in patient biopsies representing the BE-disease sequence. Differences in expression of these markers in Barrett’s patients with cancer-progression and non-progression were examined. Proliferation was reduced in Barrett’s specialized intestinal metaplasia (SIM) compared with EAC (*p* < 0.035). Correcting for cell proliferation levels, a confounding factor, linked to oxidative stress profiles, SIM demonstrated increased levels of 8-oxo-dG and 4-HNE (*p* < 0.05) compared with EAC. Longitudinal analysis of Barrett’s patients demonstrated decreased levels of 8-oxo-dG in SIM cancer progression (*p* < 0.05). BE is an environment of increased oxidative stress and inflammation. Patients with progressive disease demonstrated reduced oxidative stress levels in 8-oxo-dG. Perhaps these alterations facilitate Barrett’s progression, whereas in non-progressive disease, cells follow the rules of increased oxidative stress ultimately triggers cell apoptosis, thereby preventing propagation and survival.

## 1. Introduction

The incidence of Barrett’s esophagus (BE) and esophageal adenocarcinoma (EAC) is significantly on the rise [1,2,3]. Gastroesophageal reflux disease, specialized intestinal metaplasia of BE (SIM), low- (LGD) and high-grade dysplasia (HGD), and EAC, represents a paradigm model of inflammation-associated cancer [4,5]. Surveillance programs attempt to ensure early cancer detection and improve outcome and survival [6,7]. However, despite the increasing incidence of EAC, data indicate the cancer risk associated with BE is not as high as originally documented, and for many patients, the lifetime risk of cancer progression is as low as 0.12% to 0.5% per annum [8,9,10]. As such, the cost-effectiveness of surveying a non-dysplastic Barrett’s cohort is debatable. The clear challenge for translational science is to both understand the drivers of tumorigenesis and to also find biomarkers with a high sensitivity and specificity for identifying BE patients at greatest risk of cancer progression. 

Currently, no single biomarker is available for use in clinical practice [11,12]. This highlights the difficulty with the BE model for biomarker identification. In fact, the Northern Ireland BE Register [13] examined archived patient samples in one of the larger studies and concluded that the sensitivities and specificities of the individual biomarkers examined were low. While the Seattle Barrett’s Esophagus Study group has identified clonal diversity as a biomarker for BE progression [14,15], no biomarker has yet advanced to a Phase 5 study for development in clinical practice [11,12].

In inflammation to cancer pathways, oxidative stress and altered energy metabolism have attracted increased attention, with a particular focus on the mitochondria [16,17]. Mitochondria are the major cellular sources of reactive oxygen species (ROS) and excessive production has been linked to increased mutagenesis [18,19,20]. Mitochondrial deoxyribonucleic acid (DNA) is more vulnerable to oxidative injury and the accumulation of random mutations compared to nuclear DNA [21,22], due to the DNA’s close proximity to ROS production sites (i.e., the mitochondria themselves) and the reduced proficiency of mitochondrial DNA repair mechanisms. In fact, our group has previously demonstrated that increased mitochondrial instability—represented by increased ROS, random mitochondrial deletions and expression of the oxidative stress gene, cytoglobin—are all early events in the Barrett’s disease sequence [23]. Thus, as an extension of our previous works, we decided to explore the role of oxidative stress in relation to BE and its progression.

Studies have shown that increased oxidative stress is a precursor for increased rates of mitochondrial mutations [18]. It is well accepted that increased mutagenesis carries an inherent predisposition for cancer development [24]. While the exact pathogenesis of BE is unknown, one theory which has been implicated is bile reflux, with Jenkins et al. [25,26] demonstrating deoxycholic acid (DCA) as the most genotoxic bile acid, through pathways inducing increased release of ROS in esophageal cancer cells. In rat models, with enhanced gastro-duodenal reflux, groups have shown increased markers of oxidative stress were linked to BE change, indicating this may play a role in esophageal carcinoma [27,28]. Studies have demonstrated increased oxidative stress is a precursor for increased rates of mitochondrial mutations, with one study reporting increased random mitochondrial mutations in SIM biopsies compared to surrounding normal tissue [29]. However, human tissue studies are limited with respect to the role oxidative stress profiles may play in the actual progression of BE to cancer.

Our group has previously demonstrated a strong relationship between mitochondrial mutations and oxidative stress in the inflammatory condition, rheumatoid arthritis, using 8-oxo-7, 8-dihydro-2’-deoxyguanine (8-oxo-dG) and 4-hydroxy-2-nonenal (4-HNE) [30,31]. 8-oxo-dG is a marker of oxidative stress formed in the presence of excessive ROS production and associated with increased levels of random mitochondrial point mutations and inflammation [30]. Excessive production of ROS can result in formation of DNA adducts such as 8-oxo-dG formed by the reaction of hydroxyl (OH) radicals with the DNA guanine base. 8-oxo-dG is a pro-mutagenic lesion that mispairs with adenine, leading to GC to TA transversions [32]. This form of oxidative stress has been shown to occur in esophageal tissue following treatment with bile acid and low pH, both composites of reflux and recognized precursors of BE [33]. ROS are also lipid-peroxidation-inducing agents and 4-HNE is formed during lipid peroxidation of 6-polyunsaturated fatty acids by superoxide. This α,β-unsaturated aldehyde is highly reactive with proteins and DNA and induces cyclooxygenase-2, which controls prostaglandin production during inflammation, and interferes with nuclear factor-kappa beta (NF-κB) signalling pathways, whose activation is associated with inflammation and oxidative stress [34,35,36]. Mitochondrial proteins are targets of 4-HNE adduct formation and may influence mitochondrial uncoupling, transport and pore functions [37,38,39].

With respect to BE, this study examined the levels of oxidative stress markers, 8-oxo-dG and 4-HNE, across the Barrett’s disease sequence. We then aimed to determine if there were differences in these markers between patients who progressed to HGD/EAC compared to BE non-progressors. 

## 2. Results

### 2.1. Differences in Oxidative Stress across the BE-Disease Sequence

#### 2.1.1. Aged-Matched Histology Groups

To enable comparative analysis in the levels of oxidative stress between different histology groups representing the Barrett’s esophagus disease sequence, all groups were age-matched (*p* = 0.117). The median age of the overall patient cohort was 62 years (range 29–84 years) in a 3:2 male to female ratio.

#### 2.1.2. 8-oxo-dG Expression along the BE Disease Sequence

Representative images of 8-oxo-dG staining in BE and EAC are shown in Figure 1, panels A and B, respectively. The percent of cells positive for 8-oxo-dG in the stroma cytoplasm were significantly increased in EAC compared with normal (*p* = 0.016) and HGD (*p* = 0.035) tissue (Figure 1). There was no significant difference in positivity for 8-oxo-dG in the stroma nuclei (*p* = 0.741), epithelial cytoplasm (*p* = 0.633) and epithelial nuclei (*p* = 0.573). No difference was demonstrated in the staining intensity of 8-oxo-dG in the stroma cytoplasm (*p* = 0.910), stroma nuclei (*p* = 0.692), epithelial cytoplasm (*p* = 0.262) and epithelial nuclei (*p* = 0.569) across the BE sequence. 

#### 2.1.3. 4-HNE Expression Along the BE Disease Sequence

Representative images of 4-HNE staining in BE and EAC are shown in Figure 2, panels A and B, respectively. No significant difference in 4-HNE staining intensity was demonstrated in stroma cytoplasm (*p* = 0.646), stroma nuclei (*p* = 0.924), epithelial cytoplasm (*p* = 0.790) and epithelial nuclei (*p* = 0.653). Figure 2 shows significantly increased 4-HNE percent positivity in EAC stroma nuclei compared with LGD (*p* = 0.035). Similarly, 4-HNE percent positivity was significantly increased in EAC epithelial cytoplasm compared with SIM (*p* = 0.004), LGD (*p* = 0.003) and HGD (*p* = 0.003) samples, and also in EAC epithelial nuclei compared with LGD (*p* = 0.041) and HGD (*p* = 0.022). 

#### 2.1.4. Ki67 Expression along the BE Disease Sequence

Next, we decided to determine if there were differences in proliferation across this patient cohort. Representative images of Ki67 staining in BE and EAC are shown in Figure 3, panels A and B, respectively. Acknowledging such proliferative differences could affect oxidative stress levels, we measured Ki67 percentage positivity in the TMAs. Ki67 was 3.1-fold higher in EAC stroma compared with SIM stroma (*p* = 0.035), and 1.6-fold higher in EAC epithelium compared with SIM epithelium (*p* = 0.012) (Figure 3). No significant differences were demonstrated in proliferation levels between the remaining histological groups.

Given differences in proliferation were a potential confounding factor, we then normalized all our oxidative stress measurements across the BE sequence with our Ki67 results to better compare the overall oxidative stress levels between SIM versus EAC.

#### 2.1.5. Normalization of 8-oxo-dG Expression to Proliferation in SIM versus EAC 

Correcting for proliferation differences between SIM and EAC (Figure 4), we demonstrated significantly increased 8-oxo-dG percentage positivity in SIM stroma cytoplasm (*p* = 0.010) and stroma nuclei (*p* < 0.0001) compared with EAC. There was a significant increase in epithelial cytoplasm percentage positivity in SIM compared with EAC (*p* = 0.009). 

#### 2.1.6. Normalization of 4-HNE Expression to Proliferation in SIM versus EAC 

Following normalization of 4-HNE results (Figure 5), there was a significant increase in 4-HNE levels in SIM stroma cytoplasm (*p* < 0.0001) and stroma nuclei (*p* = 0.007) compared with EAC. No significant differences were seen in the epithelium of the EAC and SIM patients.

### 2.2. Differences in Proliferation (Ki67), Oxidative Stress (8-oxo-dG) and Lipoperoxidation (4-HNE) between SIM Progressors and non-Progressors

#### 2.2.1. Demographics of SIM Progressors and Non-Progressors 

Focussing on patients with a primary diagnosis of SIM on their first surveillance endoscopy; progressors (*n =* 13) and non-progressors (*n =* 10) were separated, with the primary end-point being progression to HGD and/or EAC. The median age of SIM patients was 59 years (26–84 years), with a 2.4-fold male predominance. No difference in age (*p* = 0.8504) nor gender (*p* = 0.873) was seen between progressors and non-progressors. The median time of progression to HGD/EAC was 2 years (Range 1–16 years). The non-progressor group was followed for a median of 7.5 years (Range 2–17 years) and had no evidence of progression. 

#### 2.2.2. Analysis of 8-oxo-dG in SIM Progressors versus Non-Progressors

8-oxo-dG staining intensity in stroma cytoplasm was significantly weaker in patients progressing to HGD/EAC (85%, *n =* 11/13) compared to 20% (*n =* 2/10) of non-progressors (*p* = 0.0075). (Figure 6A). Similarly, 8-oxo-dG staining intensity in stroma nuclei was significantly weaker (38%, *n =* 5/13) in progressing HGD/EAC compared to non-progressing SIM (0% of cases) (*p* = 0.0209). (Figure 6C). 8-oxo-dG positivity in stroma nuclei was significantly decreased in patients with progressive disease (mean 29.5%, SEM 7.3) compared with non-progressors (mean 55.0%, SEM 8.6) (*p* = 0.039) (Figure 6D). There was a significantly lower percentage of positive epithelial cytoplasm for 8-oxo-dG in progressive SIM (mean 35.6%, SEM 8.8) compared with non-progressors (mean 68%, SEM 9.1) (*p* = 0.0295). This data highlights that progressive SIM was associated with significantly reduced levels of the oxidative stress marker, 8-oxo-dG, compared to cases of non-progressing Barrett’s SIM, where staining intensity was, overall, significantly stronger, with a significant increase in positive cells. 

#### 2.2.3. Analysis of 4-HNE in SIM Progressors versus Non-Progressors

In contrast to 8-oxo-dG profiles, no difference in 4-HNE levels was demonstrated in stroma cytoplasm (*p* = 0.141), stroma nuclei (*p* = 0.540), epithelial cytoplasm (*p* = 0.399) and epithelial nuclei (*p* = 0.421) 4-HNE staining intensity between SIM progressors and non-progressors. No difference was demonstrated in the percentage of stroma cytoplasm (*p* = 0.079), stroma nuclei (*p* = 0.630), epithelial cytoplasm (*p* = 0.520) and epithelial nuclei (0.923) staining positive for 4-HNE between progressors and non-progressors. 

#### 2.2.4. Ki67 Expression in SIM Progressors versus Non-Progressors

No difference was demonstrated in Ki67 stroma expression between progressive (mean 3.3%, SEM 1.67%) and non-progressive (mean 5.2%, SEM 1.63%) patients (*p* = 0.278). There was no significant difference in Ki67 epithelial expression between progressive (mean 24.4%, SEM 7.04%) and non-progressive (mean 40.5%, SEM 8.83%) patients (*p* = 0.270). 

## 3. Discussion

The role of oxidative stress as a predictor of progression in the chronic inflammatory pathway of Barrett’s esophagus is unknown. Oxidative stress is known to play a key function in other chronic inflammatory states [30,31], yet, its involvement in BE [40] remains to be elucidated. Using archived tissue samples from patients undergoing a one-off endoscopy (i.e., normal patient biopsies) or Barrett’s esophagus surveillance, we measured oxidative stress markers in an attempt to identify differences between patients with progressive and non-progressive disease.

Similarly, we used archived tissue samples, like those in the Northern Ireland Barrett’s esophagus study. Their study examined LGD, DNA ploidy and *Aspergillus oryzae* lectin as potential predictors of Barrett’s cancer progression [13]. As testament to the difficulties with this task, their large population study highlighted that the sensitivities and specificities of these individual biomarkers were low, and as such, their clinical application has not yet transpired [11,12]. 

In our study, we examined the potential role of oxidative stress markers, 8-oxo-dG and 4-HNE as potential markers for Barrett’s progression. We chose 8-oxo-dG, a pro-mutagenic lesion, and a marker of oxidative stress formed in the presence of excessive ROS production and associated with increased levels of random mitochondrial point mutations and inflammation [30,32]. Such oxidative stress has been demonstrated in esophageal tissue following treatment with bile acid and low pH, both composites of reflux and recognized precursors of BE [33]. Aside from lower levels of 8-oxo-dG in normal stroma cytoplasm compared with EAC, initial analysis did not demonstrate significant differences in 8-oxo-dG across the BE-disease sequence. However, considering that increased cellular proliferation may influence 8-oxo-dG levels, results were normalized to Ki67 values to eliminate proliferation as a confounding variable. Like many other studies, proliferation was significantly increased in the invasive adenocarcinoma compared with SIM [41]. Adjustments for these confounding differences demonstrated increased levels of 8-oxo-dG in SIM compared with EAC. These differences were seen in both stroma and epithelial cells. It has been hypothesized that factors secreted from stroma cells in BE could influence the biology of the epithelial cells [40]; however, this warrants further testing. 

4-HNE, another marker of oxidative stress is formed during lipid peroxidation of 6-polyunsaturated fatty acids by superoxide [39]. Mitochondrial membrane lipids are extremely vulnerable to oxidative injury and lipid peroxidation can result in mitochondrial instability [39]. 4-HNE has also been shown to induce intercellular production of ROS [42], and it too has been correlated with increased inflammation and mitochondrial mutations [30,43]. Similarly to 8-oxo-dG, we demonstrated 4-HNE was significantly increased in SIM compared with EAC, highlighting BE is an environment of increased oxidative stress. In one of the few studies to examine the role of oxidative stress in BE, Lee et al. demonstrated SIM as an environment of increased mitochondrial instability compared with its surrounding normal tissue, however, their study did not examine the entire disease-sequence [29]. Like this study, we interpreted these results to mean that increased oxidative stress may be a possible driver of cancer development.

The second end-point of our study, aimed to identify differences between progressive and non-progressive disease at the earliest stages of BE and determine if oxidative stress profiles could play a role in this segregation. Noting that the environment of oxidative stress altered across the BE-sequence, we set out to examine differences in these variables between patients with confirmed HGD/cancer progression and static disease. In progressors, 8-oxo-dG was significantly decreased, suggesting that cancer progressors have an environment of reduced oxidative stress. This finding that oxidative stress was decreased in SIM with greater malignant potential, appeared counterintuitive. An explanation may be provided in a study by Trifunovic et al. on mitochondria and ageing, where increased oxidative stress, in the form of elevated ROS, was associated with mitochondrial mutagenesis in the control setting. However, in an environment of enhanced mutagenesis, the knock-on effect was disturbed respiratory chain function, defective oxidative phosphorylation and no subsequent increase in the production of ROS [18,44]. Our result also further highlights that the burst in oxidative stress levels may be more influential in driving the early stages of neoplastic transformation and becomes more redundant in terms of its action once the tumour forms. In fact, the Warburg effect theorizes cancer cells reprogram their energy metabolism, reducing oxidative phosphorylation and ROS production, potentially decreasing injury to mitochondrial DNA [45,46]. In this context, the lack of mitochondrial disruption may be a precursor for cancer development, as the normal mitochondrial response, including increased apoptosis, is inhibited, and even at the earliest stages of BE, progressors are displaying similarities to the HGD/EAC phenotype. Thus, we would suggest, and this requires further study, that progressive disease may be associated with defunct oxidative phosphorylation and is likely associated with reduced levels of 8-oxo-dG and other oxidative stress markers. 

We acknowledge certain limitations of this work. Firstly, our samples size is small. However, our findings of increased oxidative stress in SIM are supported by results from other studies [29,40,47]. One of the pitfalls with BE research is the small nature of the biopsies, which are often less than 2 mm in maximum dimension. Barrett’s diagnostic and surveillance endoscopy is an invasive procedure, which is time consuming, given the guidelines recommend targeted biopsy of suspected Barrett’s lesions together with four quadrant biopsies at 1–2 cm intervals in the entire BE segment [12]. In addition, the biopsy in its entirety tends not to contain the pathology of interest, whether this be SIM, LGD, HGD or EAC, with a single focus of disease often seen. To overcome this, we performed careful microdissection on the biopsy specimens, to ensure the correct pathology was being examined. This further reduced the size of our tissue in the TMAs to 0.6 mm cores. The small nature of these research specimens did limit the amount of concurrent immunohistochemical staining which could be performed on our TMA set. However, the main priority was to ensure the quality of the different Barrett’s pathological groups, and to rigorously test that these were not lost on deeper sections. Reevaluation of deeper sections was performed by our Pathologist to ensure the Barrett’s tissue of interest was preserved; however, where relevant tissue was lost, sample size was adjusted accordingly, with statistical analysis corrected to reflect this. Therefore, while acknowledging the small size of our study, it does highlight the dilemmas faced in BE research and the need for International collaboration. We are, however, satisfied with the quality of the tissue evaluated along our BE disease sequence. 

Another difficulty encountered with studying the Barrett’s model, is that overall rates of progression to HGD and EAC are very low [8]. As such, this hinders prospective studies and attempts to identify biomarkers of disease progression, when the majority of surveillance patients will never progress. In order to overcome this, we used archived paraffin-embedded diagnostic and surveillance samples, where we had access to follow-up data, and were subsequently able to stratify our patients into those with stable SIM and progressive disease. Importantly, first-time surveillance biopsies performed at St. James’s Hospital were used for all SIM samples in order to provide a consistent baseline and overcome the different times to progression seen between patients. The patients in the non-progressive group did not progress to HGD and/or EAC after a median follow-up of 7.5 years, thereby satisfying the criteria for non-progressive disease, given recent findings by Hvid-Jensen et al. in the largest national BE study to date [8]. Analysing 11,028 patients with histologically confirmed Barrett’s, they showed that greater than two-thirds of Barrett’s-cancers were diagnosed within the first year of follow-up. 

In summary, we have demonstrated that BE is an environment of increased oxidative stress. Progressing SIM demonstrated significantly reduced 8-oxo-dG. This may be representative of defunct respiratory function in the Barrett’s progressors, and may aid in the differentiation of those with greatest malignant potential. The mitochondrial environment may play a key role in fuelling progression, and mechanisms which may be controlling these pathways may represent a target for further study.

## 4. Materials and Methods

### 4.1. Patient Samples

Data were collected from a national BE registry at St. James’s Hospital, Dublin [48]. All research was carried out in accordance with the Declaration of Helsinki, with all patients providing informed written consent, and approval for this study was granted by the St James’s Hospital and Adelaide, Meath and National Children’s Hospital Institutional Review Board. Retrospectively, archived biopsy samples were used to construct the BE surveillance tissue microarrays (TMAs). TMAs were constructed by our biobank manager (RF) following selection of appropriate, aged-matched cases from the Barrett’s database. Tissue was obtained from patients undergoing upper gastrointestinal endoscopy with no identified endoscopic abnormality and the availability of a normal distal esophageal biopsy (*n =* 7), which acted as controls. Following an initial diagnostic biopsy, follow-up biopsies were available for all cases of SIM (*n =* 26 patients) and LGD (*n =* 13), which either remained stable, regressed or progressed to HGD or EAC during the surveillance program. 

### 4.2. TMA Construction

Haematoxylin and eosin (H&E)-stained slides from formalin-fixed, paraffin-embedded tissue blocks were used to identify specific areas of normal squamous epithelium, SIM, LGD, HGD and EAC. The areas of interest were marked by a Pathologist (B.D.) and 0.6 mm cores were taken from the blocks and TMAs were constructed. Several representative cores (mean 2, range 1 to 6) were taken from diagnostic biopsies to construct the TMAs. 4 µm sections were placed onto Superfrost Plus poly-L-lysine coated glass slides (Thermo Fisher Scientific, Waltham, MA, USA), and baked overnight at 37 °C in a tissue-drying oven (Binder, Tuttlingen, Germany). Slides were then stored at 4 °C until stained.

Diagnoses of SIM, LGD, HGD and EAC were all previously made by a specialist upper gastrointestinal consultant pathologist. All selected samples were subsequently marked and reviewed on two separate occasions (pre- and post-TMA construction) by the Gastrointestinal Pathologist (BD), in order to ensure the accuracy of the histology samples selected. In addition, as sections were taken deeper into these blocks and the H&E slides were further examined by the pathologist. In some incidences, the pathology of interest cut-out on deeper levels, and following re-examination of H&E and immunohistochemically stained slides, these cores required removal from statistical analysis, thus affecting sample size. 

### 4.3. 8-oxo-dG and 4-HNE Immunohistochemistry

A mouse anti-8-oxo-dG monoclonal antibody (Genox, Maryland, USA) was used to stain for 8-oxo-dG, a marker of oxidative stress. A mouse anti-4-HNE monoclonal antibody (Genox, Maryland, USA) was used to stain for 4-HNE, a marker of lipid peroxidation. Antigen retrieval was carried out using Triology^TM^ (Cell Marque^TM^ Corporation, Rocklin, CA, USA), which combines three pre-treatment steps: deparaffinisation, rehydration and unmasking. Sections were incubated in Triology^TM^ (1/20 dilution in distilled water) in a Princess DYB350 programmable pressure cooker on a low pressure for 10 mins. Vectastain Elite kits (Vector Labs, Burlingame, CA, USA) were used for all immunohistochemical staining. Tissue sections were intubated in 3% hydrogen peroxide (H_2_O_2_) in methanol (Sigma-Aldrich, St. Louis, MO, USA) for 30 min to quench endogenous peroxidase activity. Sections were washed three times for 5 min each in phosphate buffered saline (PBS) and blocked for 30 min with a 1:66 dilution of normal serum. Sections were incubated in primary antibody (1:40 for 8-oxo-dG and 1:40 for 4-HNE) for 2 h at room temperature. A control slide (full-face section) was incubated for the same time with PBS and no antibody. Sections were washed three times for 5 min each in PBS. Sections were incubated for 30 min in a 1:400 dilution of biotinylated secondary antibody and washed again three times for 5 min each in PBS. Sections were incubated for 30 min in avidin–biotin complex reagent, followed by three washes for 5 min each in PBS, followed be incubation for 1–5 min (depending on the level of protein expression in the tissue) in the dark in diaminobenzidine (DAB) peroxidase (Sigma-Aldrich) solution. Sections were rinsed in tap water and counterstained in Harris’s haematoxylin (Sigma-Aldrich) for 30 s. Sections were placed in a PBS bath for 5 min and subsequently rinsed in gently running tap water for 5 min. Sections were dipped in two separate baths of 100% methanol (Sigma-Aldrich) (up and down 10 times), then transferred into two separate baths of xylene (Sigma-Aldrich) for 5 min each, before being placed in a bath of xylene overnight. Coverslips were mounted using dibutylphthalate polystyrene xylene (DPX) mountant (BDH Ltd., Dorset, UK) and left to dry in a fume hood. Images were then taken using an Aperio Scanscope XT digital scanner.

Immunohistochemistry was assessed at 40X magnification in a semi-quantitative manner for 8-oxo-dG and 4-HNE by two observers (N.J.O.F. and J.J.P.), while a third observer (JOS) scored samples where differences between the two observers were noted; all were blinded to clinical outcome during scoring. The mean score was calculated for each TMA core and used for all further analysis. 

For 8-oxo-dG and 4-HNE, epithelial and stroma cells were assessed for percentage of nuclear and cytoplasmic cells with positive staining and the associated intensity of nuclear and cytoplasmic staining. The epithelial component referred to either normal esophageal squamous epithelium or the glandular epithelial cells of SIM, dysplastic BE or esophageal adenocarcinoma. The stroma, an extracellular matrix and supportive framework for the mucosal epithelium, is a mix of collagen fibres and capillaries, and composed of a variable mix of fibroblasts, endothelial cells and inflammatory cells. For the purpose of this study, the stroma cells referred to this entire cellular content. Percentage positivity was graded in the range of 0%, 10%, 25%, 50%, 75% or 100%. Intensity was graded as 0 (negative), 1 (weak), 2 (moderate) and 3 (strong). In constructing the TMAs, several representative cores (mean 2, range 1 to 6) of the pathologic site of interest were taken from >1 biopsies obtained on the days of diagnostic or surveillance endoscopy. The cores with maximum intensity and maximum percentage positivity were analyzed, as it was felt, selecting the site of maximum expression of a particular marker was more representative. The heterogeneity of Barrett’s esophagus lesions was reflected by variation in the intensity and percentage positivity from the same patient biopsies, taken from areas with the same confirmed histology at the same surveillance procedure. Following consultation with statisticians, it was realized that this variation was affecting mean values, with the mean value also dependent on the number of available biopsies taken on the day of the surveillance endoscopy, thus for consistency, the maximum values were analyzed. 

### 4.4. Ki67 Immunohistochemistry

Ki67 (Dako, Glostrup, Denmark) immunohistochemistry staining was performed using a Bond III automated immunostainer (Leica Microsystems, Wetzlar, Germany). Sections of 4µm were loaded onto the system and automated staining was carried out using Ki67 antibody. Antibody staining was detected using DAB solution and sections were counterstained lightly with haematoxylin. Coverslips were mounted onto the slides using DPX mountant. Images were taken using Aperio Scanscope XT digital scanner. The percentage of epithelial and stroma cells with positive staining were then quantified. Each core was scored by two blinded observers (NJOF and SLP), with a third observer (JOS) scoring samples where differences between the two observers were noted; all were blinded to clinical outcome during scoring. The mean value of these scores were calculated for all cores in the TMA. Once again, as several representative cores (mean 2, range 1 to 6) of the pathologic site of interest, were taken from >1 biopsies obtained on the days of endoscopy, the cores with maximum Ki67 staining were used in all statistical analysis for each patient visit. 

### 4.5. Statistical Analysis

Statistical analysis was performed using SPSS^®^ (version 18.0) software (SPSS, Chicago, IL, USA). Differences between continuous variables in the different histological groups were calculated using Mann–Whitney U and Kruskal–Wallis tests. Differences between categorical variables were analysed using Chi-squares tests. Intensity differences between Barrett’s SIM progressors and non-progressors were calculated using Chi-square tests, while Mann–Whitney U tests were used to determine differences in percentage positivity. Statistical significance was defined by *p* ≤ 0.05, and variation expressed as standard error of the mean (SEM). 

## Figures and Tables

**Figure 1 ijms-20-04449-f001:**
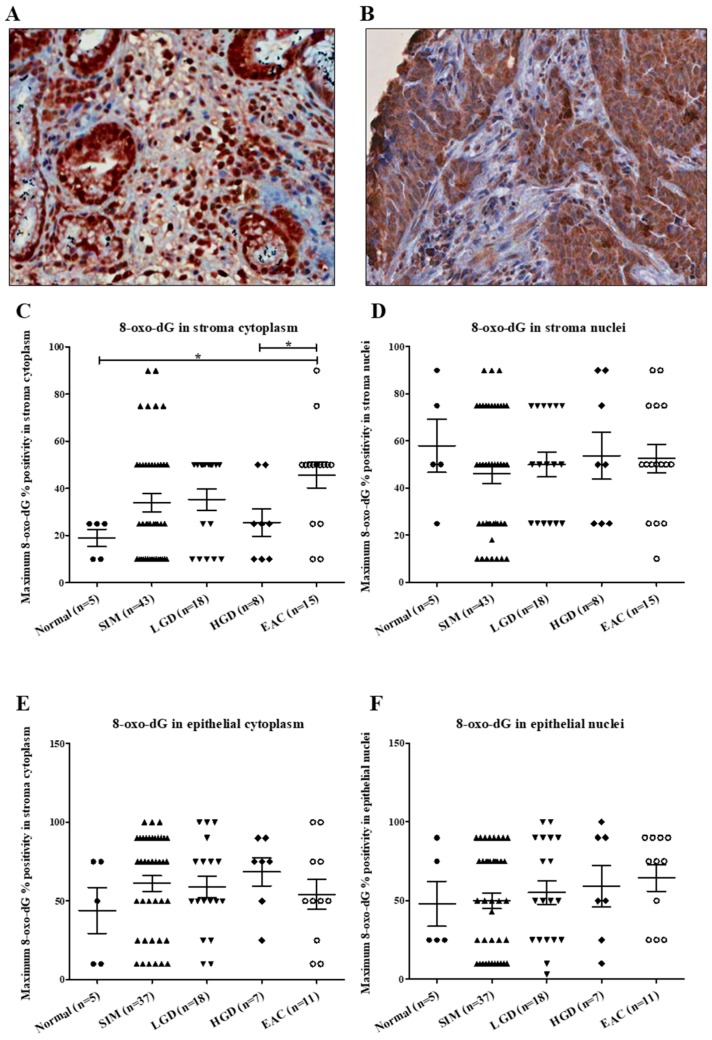
(**A**–**G**): 8-oxo-dG stroma and epithelial staining in aged-matched histology groups across the Barrett’s esophagus disease spectrum. (**A**) Section of a core of Barrett’s intestinal metaplasia, at magnification 40×, demonstrating strong nuclear and cytoplasmic 8-oxo-dG staining in 100% of the glandular epithelium. (**B**) Section of a core of esophageal adenocarcinoma, at magnification 40×, showing moderate cytoplasmic staining in 100% of the epithelium, and weak to moderate cytoplasmic staining in 50% of the stroma cells and no nuclear staining. (**C**) Kruskal–Wallis analysis with Dunn’s multiple-comparison test demonstrated a significant increase in stroma cytoplasm positive for 8-oxo-dG in esophageal adenocarcinoma (EAC) compared with normal and high-grade dysplasia (HGD) patients. No significant difference was seen in 8-oxo-dG positivity in (**D**) stroma nuclei (*p* = 0.741), (**E**) epithelial cytoplasm (*p* = 0.763) and (**F**) epithelial nuclei (*p* = 0.697). * *p* < 0.05.

**Figure 2 ijms-20-04449-f002:**
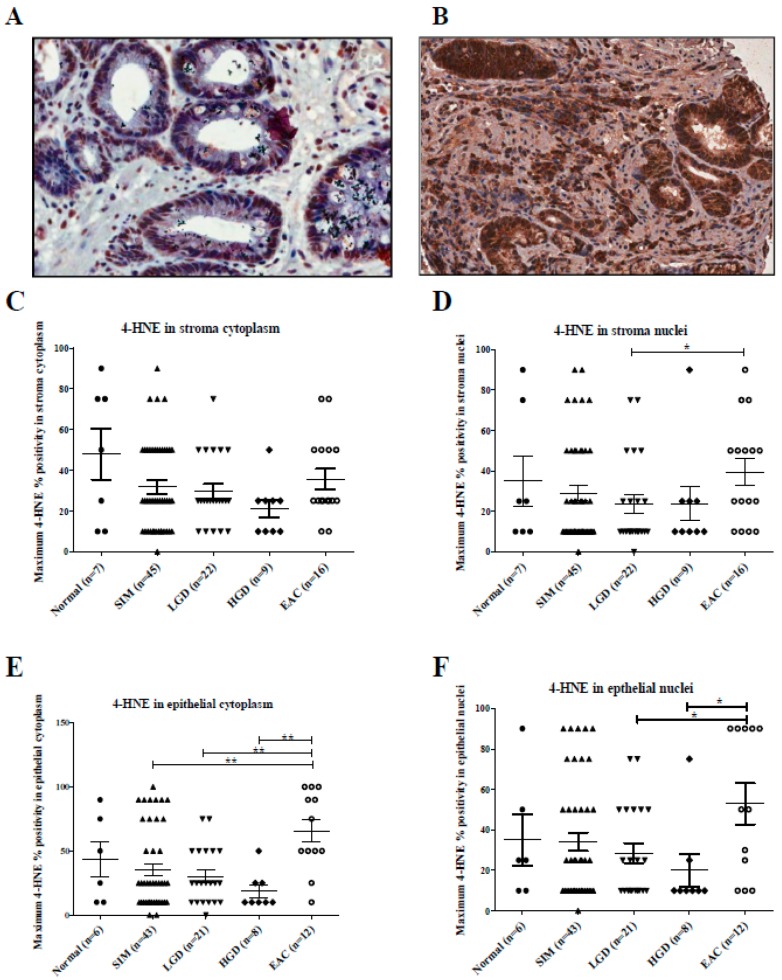
(**A**–**F**): 4-HNE stroma and epithelial staining in aged-matched histology groups across the Barrett’s esophagus disease spectrum. (**A**) Section of a core of Barrett’s intestinal metaplasia, at magnification 40×, showing approximately 50% epithelial cytoplasmic staining of weak intensity for 4-HNE and approximately 25% moderate to strong epithelial nuclear staining. (**B**) Section of a core of invasive EAC, at magnification 40×, demonstrating strong epithelial cytoplasmic staining in 100% of cells and strong staining in the stroma cytoplasm. (**C**) Kruskal–Wallis analysis with Dunn’s multiple comparison test demonstrated no significant difference in the levels of 4-HNE in the stroma cytoplasm (*p* = 0.309). (**D**) There was a significant difference in the levels of 4-HNE in stroma nuclei between low-grade dysplasia (LGD) and EAC. There were significant differences in 4-HNE levels in (**E**) the cytoplasm and (**F**) nuclei between esophageal adenocarcinoma and earlier stages of the Barrett’s disease spectrum. * *p* < 0.05, ** *p* < 0.005.

**Figure 3 ijms-20-04449-f003:**
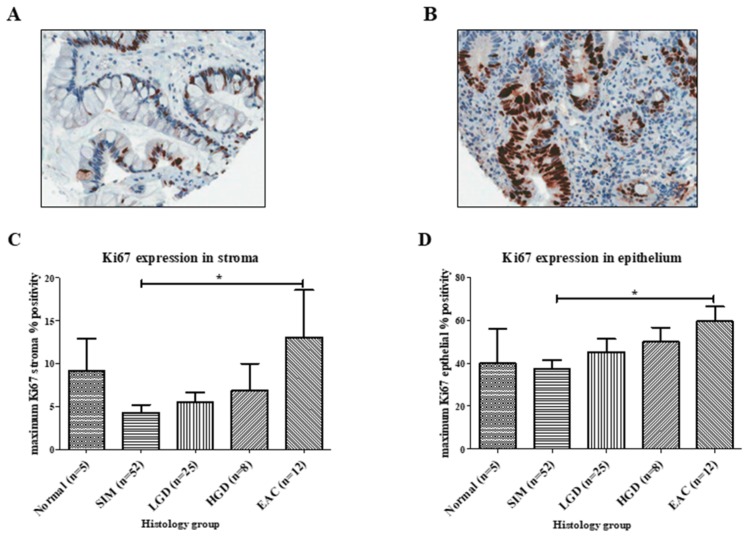
Ki67 staining across the Barrett’s esophagus disease spectrum. (**A**) Section of a core of Barrett’s intestinal metaplasia, at magnification 40×, demonstrating no staining of stroma and approximately 40% epithelial staining. (**B**) Section of a core of esophageal adenocarcinoma, at magnification 40×, demonstrating approximately 75% Ki67 staining in the glandular epithelium. Ki67 staining was significantly increased in the (**C**) stroma and (**D**) epithelial cells of esophageal adenocarcinoma patients compared with SIM patients. Error bars represent SEM. *****
*p* < 0.05.

**Figure 4 ijms-20-04449-f004:**
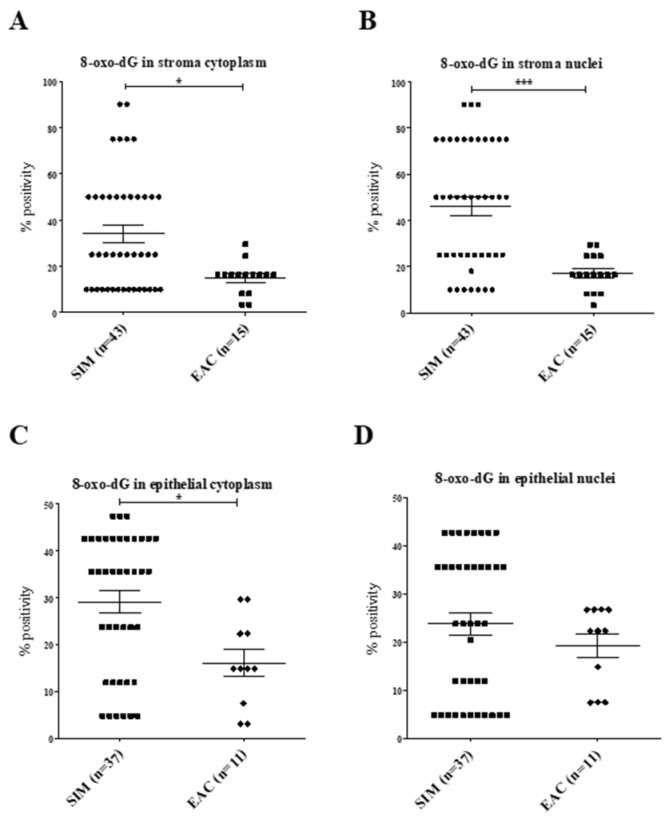
(**A**–**D**): 8-oxo-dG percentage positivity following normalization with Ki67. There were significantly increased levels of 8-oxo-dG in SIM (**A**) stroma cytoplasm, (**B**) stroma nuclei and (**C**) epithelial cytoplasm compared with EAC**. (****D**) No differences were seen in epithelial nuclei (*p* = 0.465). **p* < 0.05, ****p* < 0.0005.

**Figure 5 ijms-20-04449-f005:**
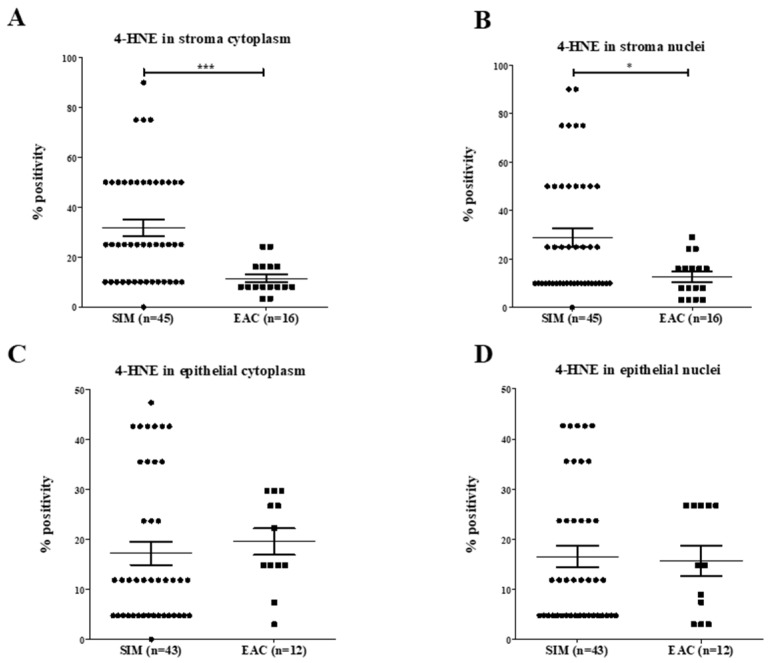
(**A**–**D**): 4-HNE percentage positivity following normalization with Ki67. There were significantly increased levels of 4-HNE in SIM stroma (**A**) cytoplasm and (**B**) nuclei compared with EAC. No differences were demonstrated in 4-HNE levels in epithelial (**C**) cytoplasm (*p* = 0.200) and (**D**) nuclei (*p* = 0.974). * *p* < 0.05, *** *p* < 0.0005.

**Figure 6 ijms-20-04449-f006:**
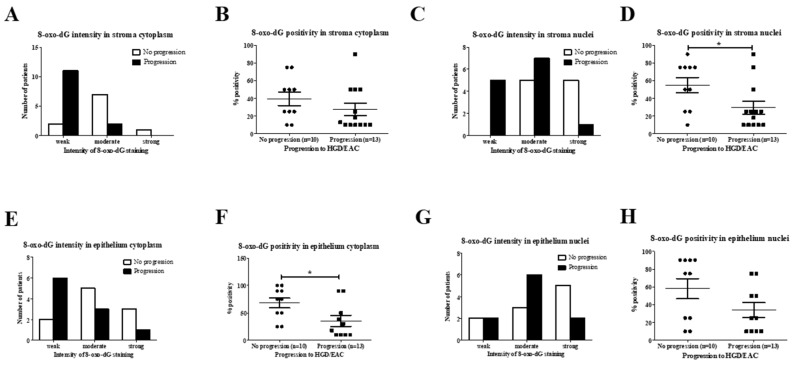
(**A**–**H**): 8-oxo-dG staining in SIM progressors and non-progressors. (**A**) Chi-square test demonstrated significantly weaker intensity of 8-oxo-dG staining in patients with progressive disease (*p* = 0.008). (**B**) Mann–Whitney U test showed no difference in the percentage of stroma cytoplasm positive for 8-oxo-dG (*p* = 0.168). (**C**) Chi-square test showed weaker 8-oxo-dG staining in the progressors (*p* = 0.0201). (**D**) Non-progressive Barrett’s esophagus (BE) was associated with an increased percentage of positive stroma nuclei (*p* = 0.039). (**E**) Chi-square test showed no difference in staining intensity in epithelial cytoplasm (*p* = 0.174). (**F**) Epithelial cytoplasm percentage positivity was significantly higher in the non-progressing group (*p* = 0.030). (**G**) Chi-square test demonstrated no difference in 8oxo-dG intensity in epithelial nuclei between both groups (*p* = 0.320). (**H**) Mann–Whitney U test showed no difference in percentage positivity of 8-oxo-dG epithelial nuclei (*p* = 0.100). * *p* < 0.05.

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
