# Peer review of "Differential Expression Profiles of Oxidative Stress Levels, 8-oxo-dG and 4-HNE, in Barrett’s Esophagus Compared to Esophageal Adenocarcinoma"

_ijms, 2019, doi:10.3390/ijms20184449_

Round 1

Reviewer 1 Report

I agree to accept the revised version.

Reviewer 2 Report

Your response to my critical comments is accepted. Thank you very much, I do not have more critical comments.

Reviewer 3 Report

I believe that now the authors have made significant changes and made it more appealing and acceptance ready.

All the changes have been correctly implemented.

This manuscript is a resubmission of an earlier submission. The following is a list of the peer review reports and author responses from that submission.

Round 1

Reviewer 1 Report

In this manuscript, Naoimh J O’Farrell et al. investigated the potential role of oxidative stress markers, 8-oxo-dG and 4-HNE as potential markers for Barrett’s progression. Their results suggested that increased oxidative stress may be a possible driver of cancer development and cancer progressors have an environment of reduced oxidative stress. The findings of this manuscript are very interesting, and well-organized and clear-presented. However, there is several concerns that the authors need to address in order to publish a well-read article.

The authors should provide the recruited patients characteristics.

Figure1 blocks its legends.

Some typo. line 177 “8oxodG ” ; line 107 “8. -oxo-dG ”  ;  line 195 “4. -HNE ” ;  line 310 “4 HNE ”.

The samples size is small, not big enough to guarantee the accuracy and reliability of the results.

In title, the authors used “profiles of oxidative stress”, but they actually just detected 8-oxo-dG and 4-HNE, so it is overstated.

Why does the authors choose 8-oxo-dG and 4-HNE? What is the connection between 8-oxo-dG and 4-HNE?

Reviewer 2 Report

In this paper, the role of oxidative stress in the sequence from Barrett’s esophagus (BE) to esophageal adenocarcinoma (EAC) has been examined in biopsies from BE-disease sequence by means of DNA adducts (8-oxo-dG), markers of lipid peroxidation (4-HNE), and cell proliferation (Ki67). This study included BE patients with cancer-progression and non-progression. In biopsies from intestinal metaplasia (SIM), the increased levels of 8-oxo-dG and 4-HNE were observed but the proliferation was reduced compared with biopsies taken from EAC patients. In BE progressing to EAC, the decreased levels of 8-oxo-dG in SIM was noticed. It is concluded that oxidative stress and inflammation are increased at BE stage not progressing to cancer, while patients with progressive disease likely demonstrated reduced oxidative stress levels. The increased oxidative stress in BE is expected to trigger cell apoptosis responsible for prevention of cell propagation and survival. 

 In overall, this is convincing evidence that oxidative stress may contribute to the pathogenesis of BE, however, few critical comments are addressed for author’s attention to clarify few issues and improve this MS in its original form.

Critical comments

The information presented in this paper are not      entirely novel. Several clinical and experimental studies with esophagointestinal      or esophagogastroduodenal anastomosis performed in the past considered      oxidative stress as the factor which might predispose or even exacerbate      the development of BE progressing to EAC. The comment on the relevant studies      referring to that topic should be mounted in the Introduction.

The major limitation of this study is number of      samples. Why only 26 patients with SIM and 14 of LGD were studied? How the      controls with different esophageal pathology criteria were matched? A      comment on that is mandatory.

Authors refer to mitochondrial ROS in      pathogenesis of BE progressing to EAC but the involvement of ROS in this      study has not been studied. Thus the conclusion that ROS could mediate these      effects is enigmatic and not supported by this data presentation.

The determination of malonyldialdehyde (MDA) and      4-HNE is considered as reliable marker of lipid peroxidation leading to      oxidative stress. Why only HNE but not the MDA + HNE were determined in      this present study?

Heterogenity of phenotypes in BE samples could be      due to different populations of inflammatory white blood cells which also      could be considered as the source of ROS. Was there any correlation of BE not      progressing or BE progressing to EAC with the density of white blood      cells? I feel that Authors should explore this point which would certainly      clarify with an attempt to correlate the oxidative stress with      inflammatory cells.

The stroma cells are not defined by this study.      Do Authors mean perhaps fibroblasts, cells of ECM or both or other cells?      This should be clarified in the text of Methods and discussed.

The contribution of stroma cells is unclear in      this study. They were found to play a role, or if no change for example in      immunostaining for Ki67, simply is considered as bystanders in BE      pathology which can be controversial.  The comment on the place of stroma cells      in BE scenario of pathology should be made in Discussion.

The final conclusion is rather the speculation on the possible involvement of apoptosis in BE-SIM. Please add the staining on apoptotic proteins Bax/Bcl2 or attach the changes in caspases system as apoptosis biomarkers in your BE samples in order to strengthen your conclusion.

Reviewer 3 Report

The study is very well oriented and organized.

The drafting is also devoid of errors however, the authors need to improve few areas in the manuscript where there are minute language errors and also the figures as they are unclear with regards to the DPI.

There are few lines like 192 and 193 which are cut and needs to be reframed.

Abbreviations in the text when presented for the first time needs to be elaborated.

In line 73 and 74, when denoting about OS and mutations, authors needs to list or add more references to support.

Also, authors need to include at least one inflammatory marker as they are stating that BE is an inflammatory disease and highlighting inflammation associated cancer.

Authors need to mention a little introduction of inflammation and its role play in BE in the introduction.